# Self-determination theory in acute child and adolescent mental health inpatient care. A qualitative exploratory study

**Josephine Stanton**[1]*, **David R. Thomas**[2], **Maarten Jarbin**[3], **Pauline MacKay**[1]

**1** Child and Family Unit, Auckland District Health Board, Auckland, New Zealand, **2** Social and Community Health, School of Population Health, University of Auckland, Auckland, New Zealand, **3** Faculty of Medicine and Health Sciences, Linkoping University, Linkoping, Sweden

* josephines@adhb.govt.nz

## Abstract

### Introduction

There is a dearth of research to guide acute adolescent mental health inpatient care. Self-determination Theory provides evidence that meeting needs for relatedness, autonomy and competence is likely to increase wellbeing and intrinsic motivation. These needs may be able to be met in the inpatient environment.

### Method

This qualitative study aimed to explore young people's experience of acute mental health inpatient care with particular attention to meeting of these three needs. Fifteen young people were interviewed. The importance of relatedness with staff, other young people and families was identified.

### Results

Relatedness with staff and peers were valued parts of admission. Some young people describe enhanced relatedness with family. They described loss of autonomy as a negative experience but appreciated opportunities to be involved in choices around their care and having more freedom. Coming into hospital was associated with loss of competence but they described building competence during the admission. Engaging in activities was experienced positively and appeared to enhance meeting of all three needs. Meeting of the three needs was associated with an experience of increased safety.

### Conclusions

Engaging young people in activities with a focus on relatedness, autonomy and competence may have specific therapeutic potential. Autonomy, experience of competence and connection with staff may enhance safety more effectively than physical containment. Peer contact may have untapped therapeutic value we understand little of. This study supports the value

**Data Availability Statement:** The ethical consent which restricts data availability was negotiated with the Health and Disability Ethics Committee. This Committee was established by the New Zealand

Ministry of Health and reviews every study in New Zealand which involves participation of people using health and disability services in New Zealand. The consent process assured participants that their confidentiality regarding their participation in the project would be maintained. Excerpts from the participant transcripts are included with the paper where they cannot be identified from the excerpts. It would not be possible to remove all potentially identifying information from the raw interview transcripts without shortening it to a point which would compromise its integrity because you can't be sure that participants could not be identified by contextual information. We have described the methods in the body of the manuscript in sufficient detail such that others could replicate the analyses in another sample population. Although data cannot be released we could provide details of the text coded into specific themes - on request. This text would be checked to ensure is has been anonymised. The non-author institutional contact to ensure ongoing data availability for this is Mary-Anne Woodnorth, manager ADHB Research Office, mwoodnorth@adhb.govt.nz.

**Funding:** A grant was awarded to Dr Josephine Stanton by the A+ Trust, URL: https://www.adhb.health.nz/health-professionals/research/ The number of the grant was:SPG-1609-002. The funders had no role in study design, data collection and analysis, decision to publish, or preparation of the manuscript.

**Competing interests:** The authors have declared that no competing interests exist.

of Self-determination Theory as a guide day to day inpatient care to meet the needs of adolescents for relatedness, autonomy and competence.

## Introduction

Acute adolescent mental health inpatient units use significant health resources to provide care to some of the most distressed and high risk young people in our society [1]. Every day these units infringe young people's liberty and there are indications that harm may occur [2–7] Harmful effects of traumatic, humiliating and coercive experiences have been documented in adult mental health inpatient care [8,9].

There is very little evidence to guide practice [4]. The significant and growing evidence base for adolescent mental health treatment is almost universally developed in community care which is quite a different context. Green's collaborative problem solving approach is a notable exception in having been directly researched in a child and adolescent inpatient care context [10].

Principles and values in trauma informed care and the recovery model are embraced by inpatient care providers. However, there is a limited evidence base and lack of clarity as to how to translate them into care [11,12]. Aspects of evidence based community based programs developed for self-harming adolescents have been adapted for inpatient use but the inpatient context is not considered ideal [13,14]. Increased calorie intake can be provided for adolescents with Anorexia Nervosa using hospital for intensive meal support or coercion but inpatient care is not demonstrated to be superior to community care [15]. Activation for depression and evidence based medication treatments for a range of disorders, particularly psychosis, in a context of containment, can be part of inpatient care.

Despite the lack of evidence to guide practice there are health gains for young people in inpatient care [16]. As with psychotherapy, it is likely that much of the improvement can be attributed to so-called "non-specific factors". In hospital these factors reside in the social and physical context which are not usually the focus of research into mental health care. These factors vary widely across institutions. There is variability in how inpatient care is accessed, length of stay and what happens in inpatient units.

This study focuses on The Child and Family Unit in Starship in Auckland NZ. It has 18 adolescent beds (intermittently used also for children) serving a population of about 2.2 million people living mostly in the main metropolitan area but also up to 6 hours' drive away. There are over twenty referring teams with funding by 8 district health boards. They expect the unit to meet the needs for admission as defined by the local community teams for assessment and treatment of mental disorders. Respite is provided locally as part of the community treatment package [17]. There are 350 to 400 admissions annually, mostly adolescents with para-suicidal behaviour or psychosis. Average length of stay is 11 days. Conduct Disorder, Substance Abuse and sequelae of trauma are common co-morbidities. Approximately 40% of admissions are involuntary. The ethnic mix includes approximately 49% NZ European, 30% Maori, 7% Pacifica, 5% Asian and 9% other.

Two of the authors (JS and PM) have been part of building a strengths-based inpatient culture minimizing control, optimizing warmth and connection and supporting the role of families in the young people's recovery. We have drawn on principles of the recovery movement [18], DBT for young people [13], the Strengths Model [19] and Johnella Bird's work [20–22].

We have felt our approach has improved the inpatient service. Recruitment and retention of nursing staff has significantly improved. The seclusion room is no longer used and has been

decommissioned. Use of restraint has also decreased. The average length of stay has reduced over 30 days to approximately 10 days. Feedback from families, by postal questionnaire, with the offer of a follow up interview, is increasingly positive but received from only 10–15% of eligible families. We wanted more depth of feedback from young people as well as families in order to increase our understanding of how they experience the care in the unit.

For the approach to be of value over time and transferable to other units it needs to be more clearly defined, ideally with an evidence and theoretical base. The central elements or our approach are reflected in Self-determination Theory [23] which has the potential to provide that theoretical and evidence base.

On the basis of extensive empirical research, Self-determination Theory has identified three basic human needs; relatedness, autonomy and competence. When these needs are met people experience increased wellbeing and intrinsic motivation. Relatedness refers to an experience of feeling connected to other individuals and a community. It includes experiences of caring for, and being cared for and accepted by others [24]. Autonomy refers to an experience of acting from one's own interest and integrated values and experiencing behaviour as an expression of the self [24]. Competence refers to an experience of feeling effective, experiencing opportunities to exercise, express and extend one's capacities and feeling a sense of confidence [24]. These three needs represent "psychological nutriments that are essential for ongoing psychological growth, integrity, and well-being" [23 p. 229]. Self-determination Theory offers the potential for empirically based theory to be guiding every aspect of practice, nursing support in activities of daily living, recreational activities, schooling, family involvement in the unit as well as clinical interventions.

The core research outcome in many of the studies is intrinsic motivation rather than recovery from mental illness and there is limited research done in mental health settings. But they have found in other settings that fulfilling these needs significantly influences a person's performance and well-being at work and school [25]. Lack of motivation is a key part of the negative symptom complex associated with limited recovery from psychotic illness [26,27]. *Developing motivation to work towards a life worth living is an essential aspect of therapy for suicidal adolescents [28]*. The Self-determination approach to understanding motivation is in stark contrast with previous theories of behaviour change which centralised reinforcement contingencies. Token economies utilising external reinforcements have been used in inpatient units in the past but have fallen out of favour [29]. In fact, external rewards such as money can reduce intrinsic motivation. People who receive an extrinsic reward for an activity are less likely to engage in that activity once the reward is no longer on offer than those who did not receive the external reward [30].

Previous research in adolescent mental health inpatient units indicates these needs are important to young people in this context and that their experiences of relatedness, autonomy and competence can be undermined by the process of being admitted to hospital. In terms of relatedness, young people have described the importance of feeling cared for and accepted by staff and peers [5,31,32]. Some described concern about disconnection from friends [5]. Supporting relationships with family was described positively [33]. Valuing autonomy was indicated by young people talking about negative feelings experienced in the context of restrictions [5,7,33] and valuing a role in decision making [33]. Young people experienced a loss of sense of competence and autonomy on admission, because of being in such an unfamiliar environment and under surveillance [5,31,33]. They appreciated the sense of staff helping them to build confidence in themselves [5] and therapeutic input to build competence, such as practical strategies like coping and problem solving skills [5,31,34,35]. Lynch et al [36] found young people's experience of warmth and support for autonomy were correlated with intrinsic motivation for treatment in a psychiatric residential treatment centre for youth.

The aim of this study is to understand the experience of inpatient care for young people and families with a specific focus on relatedness, autonomy and competence. This paper presents the data from interviews with young people.

## Method

Ethics permission for the research study was gained from the Ministry of Health Ethics Committee (ref 16/NTA/190) and local research committees in the areas where young people lived.

Eligible participants included young people aged 12–20 admitted to the unit during the study period who were fluent in English (unless an interpreter was available). Young people were initially interviewed in hospital during their acute admission. Because of their vulnerability at this time we worked with the Ethics Committee to develop a multi-step process of recruitment and informed consent. Steps were sequential and each step would need to be completed before the next could be undertaken. The steps were: (i) as part of their routine clinical assessments of the young people admitted to the unit the consultant Child Psychiatrists would assess them for competence to consent to the research. (ii) Once the young person was considered competent the clinician would ask their parents (or guardians) if they would like to meet the research interviewer (MJ) to find out about the study. (iii) The research interviewer informed the parents or guardians, both orally and with written information, about the study and sought their written consent to approach their child. (iv) The research interviewer then approached the young person and informed them about the study with written (with an information sheet fitted to their chronological age and facility with reading) and oral information. He then sought their written consent to participate in the study. The researcher was not part of the clinical team. The young person and family were assured that their decision to consent or not would not be known by the treating team and would not affect their treatment. In order to protect anonymity of participants JS and PM were not informed as to who participated and they did not see any transcript as a whole but worked from multiple extracts grouped together.

Young people were invited to face to face interviews during admission and a telephone interview 2–4 weeks after discharge. Interviews were done individually (except for one young person seen with family). Seven of the initial face to face interviews done in the unit were around 20–30 minutes with the rest ranging in length down to 5 minutes. Young people engaged in these to a variable degree. Some were difficult to engage and others talked freely. The follow up interviews were all shorter, six minutes or less and engagement was limited, with three of the young people saying little more than that they had nothing to add. These interviews generated 120 pages of transcript. Questions guiding the interview are shown in Table 1. These were written to cover the range of the inpatient experience as openly and comprehensively as possible.

## Participants

Of the 65 young people eligible for the study 15 young people participated. Thirty six of the 50 young people who did not participate had not been approached as the researcher had not been able to complete the consent process with the family. Of the other 24, 7 declined, 5 were considered too unwell and 2 did not speak English. Fourteen participants were European New Zealanders and one was Maori. Six of these 15 young people completed the follow up. Reasons for young people not completing the follow up interview were either due to (i) the young person still being admitted to the unit, (ii) the researcher not being able to get a hold of caregiver, (iii) patient not considered well enough by carer, or (iv) young person not wanting to participate.

**Table 1. Questions guiding the interview.**

- How was your time at the child and family unit?
- What have you found helpful? What have you found unhelpful?
- How did you find it arriving at the unit? What was your perception before coming here?
- How have you experienced the staff around the unit? The nurses, doctors, key workers?
- What has been your experience of the other young people at the unit?
- How do you feel about your confidence to manage your life and challenges that lie ahead?
- How have you experienced having a voice and a choice at the unit?
- How has the relationship to your family changed since being admitted?
- What do you think of the activities/groups/school/facility/food at the unit?
- How have you experienced the family meetings?
- How have you experienced the medication at the unit?
- What's been your experience of the rules? Any rules that you would like to see at the unit?
- What's been your experience of the smoking at the unit?
- How have the spiritual and cultural needs been met at the unit?
- How have any other physical health needs been addressed at the unit?
- What would you tell someone else coming to the unit about how it has been for you?
- Is there anything else you'd like to say or anything you'd like to let us know?

## Data analysis

Analysis followed the guidelines for a general inductive approach [37] which is similar to a thematic analysis [38]. Each member of the research team independently read transcripts (whole transcripts for DT and MJ and grouped transcripts for JS and PM) and developed a preliminary set of themes. These were then shared, compared and contrasted with each other and the data. JS and DT then independently (i) coded the data using NVIVO, (ii) compared codes, (iii) developed an initial summary and (iv) returned to the data to confirm, or disconfirm, the thematic analysis we were developing.

After this JS returned to the raw data and applied a template analysis approach [39], where data were analysed specifically using the three needs identified by Self-determination Theory, relatedness, autonomy and competence. The initial analysis by the whole study team, most of whom were only minimally familiar with Self-determination Theory, optimized the possibility that aspects of the data which did not resonate with Self-determination Theory could be brought forward, thus making it less likely that we were forcing the data into predetermined categories [40].

Differences were addressed through an iterative review process back and forth between codes and raw data, to reach collective agreement around key and most relevant patterns in participants' experiences. Emerging analyses were returned to MJ and PM for further checking and comparison. Disagreements between coders throughout the whole analysis process were treated as an opportunity to access further depth in our understandings by clarifying the nature of the disagreements and what they were based on. This process contributed to the development of the coding in that it brought forward different aspects of the data. We went through several iterations before settling on the final analysis.

## Results

### Overview

Young people described largely positive experiences, most of which could be understood within the framework of relatedness, autonomy and competence. They spoke of valuing relatedness with staff, peers and families. This supported their experience of safety. Autonomy, was described as limited at times and young people appreciated a sense of choice and being heard. Several young people described the process of admission undermining their experience of competence. They also described developing competence during admission. The only issues

which were not able to be analysed within the SDT framework were some mixed feelings about medication and privacy and dissatisfaction with food.

## Relatedness

Most young people reported that staff were available, positive, caring and competent. This contributed to a feeling of safety from risks of engaging in self harm or absconding. Feeling safe in the Unit was a key theme reported by several young people.

Young people described feeling genuine caring from staff. They described nurses, teachers and other staff, as "*awesome*", "*nice*", "*kind*", "*absolutely lovely*", "*compassionate*", "*good people*".

> *Everyone's so kind and always smiles at you and even the dinner man*

> *it felt like I was at home in the hospital . . . it really felt like home.*

Most young people described the sense of being in the unit as positive; inviting, friendly and fun but with some reservations.

> *I like it here but I don't wanna stay here.*

They described appreciating staff offering practical support, eg welcoming them, showing hospitality, finding them more clothes or addressing general health concerns.

> *both very professional but also still very human and kinda down to earth and understanding*

They described staff being accepting.

> *I don't have to hide anything in here, we just be who we are without worrying about what people think of us.*

They appreciated staff considering how they felt and listening without interrupting; talking without "*droning on*" about concerns, but "*laid back*" with responses in a "*conversational rather than confrontational*" way and sometimes knowing when not to talk.

> *they paid a lot of attention to you and they were always a nice shoulder*

They described observing staff supporting others; parents, peers and student nurses.

> *treat everyone like they're the most important person in the room*

They described forming bonds with the nurses through spending time and joking. One spoke of nurses as "*temporary friends*". They appreciated their positive energy, enthusiasm and availability to chat or engage in cards or other games.

> *At night sometimes we'll play cards. . . . It's good. We will all get into groups with our favourite nurses and our favourite patients and we'll play.*

Safety was a key issue identified and was connected to perception of staff availability and commitment.

*I knew that I couldn't do anything silly or stupid to hurt or do anything harmful to myself because there was people watching and caring for every move that I did.*

Another factor in feeling safe was the sense of competence of staff, who were able to help them with relationship or emotional issues, lift their mood or calm down, understand and offer little bits of wisdom, tips and advice.

*here's a bunch of people that I know I can rely on and I can have faith in to help me with stuff*

There were a few negative descriptions, most of which focused on limits to autonomy. Some young people described choosing not to engage with staff. The staff group who were most criticized were doctors.

*They just come and tell you stuff like how long you're gonna be here. They don't really do any therapy with you. . . . Just bring up things that have happened to you that you don't wanna talk about.*

## Contact with peers provided company, understanding and belonging

*Just having friends around. . . . That's probably what made it most bearable*

Participants described feeling connected to other young people, even after they left. They described other young people as "*all pretty nice*"; a general sense of companionship. They described the other young people as helpful in welcoming them and orienting them to the unit, encouraging them to join in and offering them advice and strategies.

An important part of the sense of comfort they experienced with peers was because other young people had similar experiences; they felt understood and less alone.

*You can trust them, you can care for them, knowing that they're all going through most likely something like you are.*

*being able to relate and share and just know you're not alone in your problems*

In contrast one described it as "eye opening", realizing how much worse other people's problems were than theirs. Another young person spoke of feeling out of place.

*I didn't think why I was here was as extreme as everyone else's.*

There were other negatives described; eg, feeling isolated from their peers outside the unit and finding it hard when peers left. Several young people described how peers could be annoying or invade their space. A few talked of being influenced negatively, eg, finding it hard to contain themselves when peers were "*acting up*". Another described having peers "*playing up*" also meant less staff attention.

*I just want it to be quiet, especially when you need the nurse to talk to and everyone's too busy.*

## Relatedness with family was supported

Most of the young people described feeling well connected to their families through the admission, some increasing the sense of connection of their family during the admission. Practical

support included onsite accommodation for families, ease of making phone calls, staff welcoming families with long visiting hours and encouraging outings with family. Some young people described finding the staff helped their families develop more understanding.

*I think staff's dissection of everything, 'this is what I notice about your son, this is what we've come to a conclusion of', I think that's helped both me and my parents in understanding issues.*

One person described things being more difficult as their family developed more understanding of the seriousness of what they were going through.

*Pretty awkward but that's what I expected. . . . Probably just cos when you come to hospital then everyone thinks it's serious.*

## Autonomy

### Limitations to freedom

*It's frustrating being locked up, being restricted to a lot of things . . . missing out on life, not experiencing what a normal teenager should experience.*

This was particularly marked in the locked area.

*After I got moved over to this unit, the open one, I don't know how to explain it, I just felt better about myself, that I could talk about my feelings more*

Young people liked being able to go for walks and use the kitchen to make their own food. Some described 15 minute checks as intrusive, but could accept them.
Several spoke of understanding the importance of boundaries.

*the staff are really nice but they're also strict when they need to be*

Some described benefits from not being able to have their phones at night in terms of going to bed earlier or learning other strategies to sleep. Some wanted staff to take more control over peers, in terms of protecting privacy or containing aggressive behaviour. There were, of course, objections to rules, particularly restrictions on smoking, use of phones and staying up past 9.30.
Safety was attributed to rules and boundaries by some young people, with some even advocating for more rules to increase safety. But it was more common for young people to describe support of staff as what supported safety. One person described feeling more safe on moving out of the HDU because of having more people to talk to.
Young people spoke about appreciating being enabled to move out of their comfort zone in terms of engaging in activities they would not otherwise have done.

*They don't exactly force you to do anything but they quite suggest it.*

They appreciated staff flexibility.

*There was something they were doing the other day and I didn't wanna do that but she said this whole other thing that I could do. It makes you feel good that everyone's looking out for you.*

In contrast, a few described more pressure than was helpful, either to stay in a group, or to engage in activities.

*Sometimes it's hard when you're not in the right space and (staff member)'s being all pushy and trying to get you to do something . . . I get more upset. . . . Leave me alone.*

## Having a voice and choice

Several young people described staff sharing information openly and seeking their views and taking them seriously.

*they've asked my feedback a lot and they've listened a lot to it*

In contrast, some young people described not having a choice about medication and another described being given misleading information about medication or being asked a lot of questions, at times repetitive questions, without feeling listened to.

*It feels like I'm just talking to someone that's just taking information and receiving it. They're not really giving me any feedback*

Other young people described acceptance of limited choice. Some described appreciating choice about food.

*I can't say anything that will get me released from here which is fair enough but, I've been able to say where I wanna go after this*,

Young people talked about valuing what was sorted out in meetings with family and professionals but feeling they did not "*have a say*".

*I felt like they were sorta just talking to each other and not including me or asking me how I felt about much. . . . I feel like you're not all that in control of your own life.*

Some also talked about finding it difficult to process the information in meetings in order to put forward their views. They appreciated staff flexibility in supporting them to find a context in which they could express themselves, eg in going for a walk, rather than expecting the young person to sit and talk in a room. They described appreciating staff efforts to keep them informed about meetings and flexibility in ways of accessing their views.

*They did pretty well and tried to get me into it and make them short and sharp; tried to get to the part where I come in nice and easy, short and they've already done all the meeting and the talking and the questions and they just run over what happened with me.*

## Competence

Several young people described a sense of loss of confidence and competence on coming to the unit.

*it's like when you realise 'I'm being put into a mental ward'. It's a kind of tragic feeling, really, cos it's like, this is how low I've fallen.*

This was exacerbated by the unfamiliar environment, new people and lack of understanding about the purpose of their admission. They appreciated being shown around, introduced and having a program available.

*. . . it's just good to know what's happening at what time.*

Some young people said that just by having a break in the unit they felt refreshed and more able to get back on their feet and re-engage with their lives.

*I could start fresh because I'd had a week to recollect myself . . . Less flustered and calm and clear and ready.*

There were reports of increased confidence to engage actively in making choices in their lives, having a different outlook or perspective.

*I'm gonna think through all of the choices and challenges that I make. . . . Just a lot more confident to actually say something, not just sit at the back just agreeing with everything that's given at you.*

Several participants mentioned that engaging in the activities provided in the Unit created positive experiences which enhanced their competence.

*Even just playing pool has been pretty fun. I've only just started playing it and I was so bad at it when I first started but then I got better. Then the drums, I've also learned some new drumbeats here so that's good. Playing those two things take my mind off a lot of things and I've just become more confident with myself.*

Participants spoke of having learnt strategies for sleep, managing feelings, thoughts, stress and being more able to express themselves. This learning came from a range of members of staff and peers, groups, formal therapy and informal individual conversations.

*I feel significantly more confident than I did before coming in here. I still feel emotional, sad and anxious but I feel like I've learned a lot of things and lessons here that over time I will be applying that will help me as an individual just cool off and be a generally healthier person.*

One person spoke of the limitation of learning strategies to manage feelings.

*It doesn't deal with the cause of them*

Some described getting lost in the "*Starship reality*" had undermined their competence to cope in the outside world.

*. . .it's a completely stress-less environment and you get adapted to it and you start to get used to that slow, stress free way of doing things . . . It's very hard to adapt and re-accustomise (sic) yourself with the way the world actually works beyond the hospital unit.*

A specific competence some young people would like to have seen addressed was how to manage inquiries from peers on returning home.

Some described having lost competence in schoolwork and felt the unit school helped in getting them back into routine, whereas others found it was not stretching them enough or felt that not enough was done to keep them up with their studies.

*The little tasks, it's quite simple but it gets you thinking and gets your brain working again, . . . I haven't been to school in about a year so getting back into it and doing a few little tasks is a good way to start it off.*

Positive effects of engaging in activities were often spoken of, in terms of distraction from distress and increasing positive feelings, particularly in the accepting environment.

*Yeah, I've done art. That was really quite different. It was cool, though. It was a bit like therapy, honestly. It was quite relaxing, you just got to do whatever you wanted to do without getting judged or your piece of art would get commented on in negative ways. You wouldn't judge, you wouldn't really talk about another person's piece of art or something. It was really quite relaxing and actually really fun.*

## Discussion

Young people described activities very positively. Activities are often thought of as valuable for distraction and enjoyment. If they are focused so that they meet the needs identified by Self-determination Theory it is also possible that they could have specific therapeutic value in contributing to intrinsic motivation and wellbeing. The young people described valuing engagement with a focus on encouragement without undermining autonomy and enabling experiences of competence and supporting relatedness. Activities which meet needs for autonomy, relatedness and competence could be further explored in the community context as a possibility for a therapeutic intervention for the many young people who are difficult to engage in talking or medication therapy.

Peer contact may have untapped therapeutic value we understand little of. Peer support is well established in adult mental health care with some evidence of effectiveness in community care [41,42]. Employment in peer mental health services also provides a career path for people with lived experience of severe mental illness [43]. Relatedness with peers has previouslyly been identified as a significant positive factor in the admission experience [35]. This is in contrast with the concern about contagion among young people both generally and particularly in inpatient units [44–47]. Smith-Gowling et al [46] reported young people finding exposure to self harm in peers in the inpatient context extremely challenging. They also reported young people describing learning from and copying each other as well as competing with each other with respect to self harm being genuine rather than attention seeking. However, young people are describing significant benefit in peer contact and there is a need to increase our understanding as to how to optimize this and minimize potential harm.

Autonomy, experience of competence and connection with staff may enhance safety more effectively than physical containment. Some of the descriptions the young people make of their experience question simplistic notions of containment providing safety. Containment and management of risk are frequently requested [17]. However, there is no evidence for the effectiveness of inpatient containment as a strategy for managing suicidal risk [48]. Open wards in adult mental health units in Europe have no higher rate of adverse events compared with locked ones [49]. The young people in this study described increased sense of wellbeing in an open environment and an increased sense of safety with feeling connected to staff they

trusted. It is of interest that despite this, some of them described the need for more restrictions to increase safety. It appears that the relationship between experiences of containment, autonomy and safety are complex and worthy of further study.

Self-determination Theory has not been applied as widely in mental health care as it has in fields such as education, sports coaching and personnel management. However, it has been put forward as a framework for the recovery paradigm [50,51]. It has been applied in other contexts such as enhancing motivation for people with severe mental illness to engage in exercise [52] and compulsory care for forensic patients [53].

## Clinical implications

Self-determination Theory has the potential to guide clinicians over a range of interactions, eg: (i) a nurse encouraging a young person to get out of bed, (ii) a doctor asking a young person to come to a clinical session, (iii) a key worker looking at how to support a young person in a meeting with family and clinicians or (iv) any staff member interacting in any role at any time. Consideration of how what we are doing is affecting the young person's experience of relatedness, autonomy and competence has the potential to guide how we listen to young people, tone of voice, style of speaking and attending to verbal cues.

## Strengths and limitations

In terms of general understanding of adolescents' experience of acute mental health inpatient care this study adds to a small and limited pool of data. Most previous studies are from longer stay units or residential treatment and few include young people with psychotic illness. There are only two studies of units with average length of stay less than 10 days [34,35]. This is important as the trend in inpatient care is towards shorter stays [54]. The focus on the principles of Self-determination Theory which can provide guiding principles for inpatient care has direct implications for using the results in everyday care.

Limitations include the small number of eligible patients effectively recruited for the study. Much of the loss of eligible participants was due to the short stay combined with the complexity of the recruitment process required to protect young people with the level of vulnerability they have during an admission. Maori, Pacifica and Asian patients are under-represented. The interviews are short which is to be expected in interviews with adolescents in acute inpatient mental health care. The difference in length and engagement in the interviews done face to face in the unit and follow up by phone supports the approach to interviewing young people while in hospital despite the challenges involved in a safe consent process for this vulnerable population. The richness of qualitative data is inevitably limited in such short interviews, but this exploratory study opens important avenues for further study.

The quasi-insider status of two authors as has been helpful in recruiting, and offers an increased understanding of the relevant issues, but also has the potential for bias. The outsider status of the other two authors offered a check for this, with reflective feedback taking place throughout the analytic process.

## Conclusion

Self-determination Theory may have the potential to fill a gap in the evidence base needed to guide the wide range of day to day interactions in inpatient care. Further research is needed.

## Author Contributions

**Conceptualization:** Josephine Stanton, David R. Thomas, Maarten Jarbin, Pauline MacKay.

**Data curation:** David R. Thomas, Maarten Jarbin, Pauline MacKay.

**Formal analysis:** Josephine Stanton, David R. Thomas, Maarten Jarbin, Pauline MacKay.

**Funding acquisition:** Josephine Stanton, Pauline MacKay.

**Investigation:** Josephine Stanton, David R. Thomas, Maarten Jarbin, Pauline MacKay.

**Methodology:** Josephine Stanton, David R. Thomas, Maarten Jarbin, Pauline MacKay.

**Project administration:** Josephine Stanton, Pauline MacKay.

**Resources:** Pauline MacKay.

**Supervision:** Josephine Stanton, David R. Thomas.

**Validation:** Pauline MacKay.

**Visualization:** Pauline MacKay.

**Writing – original draft:** Josephine Stanton, David R. Thomas, Maarten Jarbin, Pauline MacKay.

**Writing – review & editing:** Josephine Stanton, David R. Thomas, Maarten Jarbin, Pauline MacKay.

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
