## [Decision Letter · Decision Letter 0]

14 Feb 2020

PONE-D-19-29622

Self determination theory in acute child and adolescent mental health inpatient care. A qualitative exploratory study.

PLOS ONE

Dear Dr Stanton,

Thank you for submitting your manuscript to PLOS ONE. After careful consideration, we feel that it has merit but does not fully meet PLOS ONE’s publication criteria as it currently stands. Therefore, we invite you to submit a revised version of the manuscript that addresses the points raised during the review process.

We would appreciate receiving your revised manuscript by Mar 30 2020 11:59PM. To enhance the reproducibility of your results, we recommend that if applicable you deposit your laboratory protocols in protocols.io, where a protocol can be assigned its own identifier (DOI) such that it can be cited independently in the future. For instructions see: http://journals.plos.org/plosone/s/submission-guidelines#loc-laboratory-protocols

We look forward to receiving your revised manuscript.

Kind regards,

Janhavi Ajit Vaingankar

Academic Editor

PLOS ONE

Journal Requirements:

2. Please provide additional details regarding participant consent. In the ethics statement in the Methods and online submission information, please ensure that you have specified (1) whether consent was informed and (2) what type you obtained (for instance, written or verbal, and if verbal, how it was documented and witnessed). As the study included minors, ensure you have stated what consent you obtained from parents or guardians.

3. Thank you for explaining the restrictions on your data availability. Specifically you wrote, 'Data cannot be shared publicly because of ethcal considerations. Access to the interviews would have the risk that the participants might be able to be identified. Tneir anonymity was an important part of the ethical requirements of the study and was guaranteed as part of the information and consent process. Data sharing was not specified as part of the consent process. The participants, as young people in the process of acute mental health hospitalisation are a particularly vulnerable group and need to be respected.'

To ensure the statement is as clear as possible, please include the name of the IRB that imposed the restrictions on data sharing as part of the study approval. Also, in the statement, please discuss what data you are sharing with the paper, for instance, "Relevant excerpts from the participant transcripts are included with the paper." Finally, please ensure that you have described the methods in the body of the manuscript in sufficient detail such that others could replicate the analyses in another sample population.

Please include this information in your cover letter, and we will update the data availability statement on your behalf.

Reviewers' comments:

Reviewer's Responses to Questions

**Comments to the Author**

1. Is the manuscript technically sound, and do the data support the conclusions?

Reviewer #1: Yes

Reviewer #2: Partly

2. Has the statistical analysis been performed appropriately and rigorously? 

Reviewer #1: N/A

Reviewer #2: N/A

3. Have the authors made all data underlying the findings in their manuscript fully available?

Reviewer #1: No

Reviewer #2: Yes

4. Is the manuscript presented in an intelligible fashion and written in standard English?

Reviewer #1: Yes

Reviewer #2: Yes

5. Review Comments to the Author

Reviewer #1: Thank you for this lovely manuscript on an important topic.

Following are my minor revisions:

On line 35 and on line 557 please remove the formulation "may be more effective than talking or medication therapy". This conclusion can not be drawn by this study design and can not stand as it is.

Move from line 79 to 106 to Methods.

Add reference on line 129 after illness.

Add reference on line 131 after central.

On line 183 add median in minutes.

On line 445 change "re" to "in".

Erase: line 534 to 553 irrelevant to the topic.

Erase line 564 to 565 This offers...

Edit reference Smith-Gowling on line 572

Change on line 598 "can" to "may".

Reviewer #2: This study sheds light on the sensitive and important issue of children and adolescents in MH psychiatric facilities. As such it brings up an ongoing problem that is under addressed and sought in research. It makes an important contribution. However, the manuscript needs to be enhanced in its quality and revised.

Here are some points according to the sections of the article:

Introduction:

1.More knowledge to highlight the problem at first re the traumatic impact of hospitalization of mental health patients is needed. There has definitely been quite some work around that with adult manta health care that can be relevant to emphasize the need to address this problem.

Fror example, there are studies that specifically showed traumatization from psychiatric hospitalization at least in adults - which may be worth noting. For example see:

D. Paksarian, R. Mojtabai, R. Kotov, B. Cullen, K.L. Nugent, E.J. Bromet (2014). Perceptions of hospitalization-related trauma and treatment participation among individuals with psychotic disorders, Psychiatr Serv. 65(2): 266–269. doi: 10.1176/appi.ps.201200556

2.The authors might be interested to describe more about the role of providers and the link between non-specific factors and support for recovery in mental illness, which has been researched at least with the adult population. For example:

Moran, G.S., Mashiach-Eizenberg, M., Roe, D., Berman, Y., Shalev, A., Kaplan, Z., & Epstein, P.G. (2014). Investigating the anatomy of the helping relationship in the context of psychiatric rehabilitation: The relation between working alliance, providers’ recovery competencies and personal recovery. Psychiatry Research, 220(1-2), 592-597

For an example of a focus on enhancing person-centered approach and recovery oriented interventions see:

Hornik‐Lurie, T., Shalev, A ., Haknazar, L., Garber Epstein, P., Ziedenberg‐Rehav, L., Moran, G. S. (2018). Implementing recovery‐oriented interventions with staff in a psychiatric hospital: A mixed‐methods study. Journal of Psychiatric and Mental Health Nursing, 25(9-10), 569-581

Another earlier work has focused on eliminating restraint and seclusion in a psychiatric facility that might be of interest is:

Ashcraft, L., Anthony, W. (2008). Eliminating seclusion and restraint in recovery-oriented crisis services. Psychiatric Services 59(10):1198-202. doi: 10.1176/appi.ps.59.10.1198.

3. Please specify what DHB (and other acrynoms appearing in the text) stands for.

4. Deci & Ryan conducted their studies in a broad range of settings and populations, it would be interesting to discern if and what were those studies that referred only to mental health populations. If there aren’t such studies, then the next sentence needs to be modified:

The core research outcome in many of the studies is intrinsic motivation rather than recovery from mental illness but fulfilling these needs significantly influences a 128 person’s performance and well-being at work and school (Ryan & Deci, 2000).

As currently it misleads the reader regarding Ryan & Deci’s work as if it refers directly to recovery in mental health.

Methods:

More detail is needed.

1.For example, It is not clear when the interviews were taken and where.

2.In addition, if the interviews were conducted during the time of hospitalization, more information needs to be provided regrading ethical considerations of the person’s condition.

3.Interview guide: please state the rationale for the questions in the interview guide, and also describe the interview process (where it took place, how long on average interviews were? Etc.)

4. If possible, please describe the number of pages of transcript produced from the 15 interviewees to give a sense of the amount of the materials analyzed.

Data Analysis:

1.Please describe how incidences of disagreement between coders of the interviews were addressed in the analytic process.

2. Given that you are relying on SDT as a conceptual guide in your analysis, perhaps you would like to address “theory driven approach to qualitative studies”, and how your work addressed relevant issues. (see for example : Macfarlane A, O'Reilly-de Brún M. (2012). Using a theory-driven conceptual framework in qualitative health research. 22(5):607-18. doi: 10.1177/1049732311431898.)

Results

1.The opening statement needs to be improved to convey to the reader you are first presenting and overview of the findings, and then will provide elaborations with quotes for each theme.

2. More clarification on the use of the term “safety” as a main result that participants describe is needed. Is it safety in a physical sense? Is it in an emotional sense? Safety in relation to what? To how they felt outside the unit? To their sense of dealing with their current condition? It is unclear in the opening paragraph. Indeed it is broken down and clarified later in the result section – but some short way to summarize the different aspects in the opening will be helpful.

Discussion

*The discussion can take a further step to integrate the findings back to SDT other contemporary works in MH. Specifying how they add (mostly by addressing the younger population it seems – and perhaps there are some nuances of the NZ culture and norms that are different from other countries or perhaps similar?

*The discussion is also the place where the authors may consider demonstrating how the findings can be interpreted in light of the SDT-oriented models they’ve been using, that may have supported the positive findings. This can be very interesting to readers to learn about.

Followed are some specific points and suggestions:

1.I would be more careful in stating that:

“Engaging young people in activities with a focus on relatedness, autonomy and competence may be more effective than talking or medication therapy”.

However, perhaps one can definitely argue that in beginning stages where medication may be heavily sedative and unpleasant, or still has not reached fullest impact, the relational, SDT aspects are more important.

2.Citing how the current work adds to other attempts to veer psychiatric facilities to be more person centered may be also relevant (see for example, Ashcraft & Anthony, 2008, and there are more updated works in this area)

3.In terms of the value of peer contact, and its relation to SDT in mental health, there is growing research in the area of adult peer support that might be relevant. See for example:

Moran, G. S., Russinova, Z., Yim, J.Y., Sprauge, C. (2014). Motivations of persons with psychiatric disabilities to work in mental health peer services: A qualitative study using Self-determination theory. Journal of Occupational Rehabilitation, 24(1):32-41

Or

Moran, G. S., Russinova, Z., Gidugu, V., Yim, J.Y., Sprauge, C. (2012). Benefits and mechanisms of peer providers with mental illnesses. Qualitative Health Research, 22(3), 304-319

And other works related to peers, compeers, peer specialists etc., by Cheryl Gagne, Mark Zalser, Mathew Chinman, Steve Gillard, Mike Slade.

4. Clinical implication can perhaps also address how the models already used in your setting are support SDT, autonomy-supportive environments. Also enhancing training on person centered and goal centered interventions can further support SDT (see for example Hornik et al., 2018, and also Moran et al., Perceived assistance in pursuing personal goals and personal recovery among mental health residents across housing services. Psychiatry Research, 249, 94-101 – in it more relevant reference).

5.I would bring upfront to the opening paragraph of the discussion the part in the strengths that states:

his study adds to a small and limited pool of data. Most previous 604 studies are from longer stay units or residential treatment and few include young 605 people with psychotic illness. There are only two studies of units with average length 606 of stay less than 10 days (Grossoehme & Gerbetz, 2004; Moses, 2011). This is 607 important as the trend in inpatient care is towards shorter stays (Case, Olfson, 608 Marcus, & Siegel, 2007).”

All the best and good luck

6. PLOS authors have the option to publish the peer review history of their article (what does this mean?). If published, this will include your full peer review and any attached files.

Reviewer #1: No

Reviewer #2: Yes: Galia S. Moran

---

## [Author Response · Author response to Decision Letter 0]

1 May 2020

I found the reviewers' comments very helpful and have revised the paper in line with most of the comments they made. 

I have not been able to supply th median length of interviews in minutes.

There were some recomendations I did not agree with. 

Reviewer #1 wanted the description of the context of the study moved to methods. However, I feel the paper reads better the way it is. My description of teh context sets the scene for the study. 

With respect to reviewer #2, I have taken up most of their suggestions. However, I have not amplified the introduction and discussion as much as they were recomenging. The points which are made in the paper are very clear and focused. They are also quite specific to the utility of addressing the thee needs for autonomy, relatedness and competence. There is considerable literature available about related issues such as non-specific factors in recovery, person centred care which is very important. I have touched more on several of the issues they raised but kept this very limited as I beleive that the shorter any piece of writing is, the more likely it is to be read. I would be concerned that the simple and important message about the potential usefulness of Self-determination Theory in the inpatient (and potentially outpatient) context could be lost by too much spreading of the discussion around broader issues

---

## [Decision Letter · Decision Letter 1]

3 Jul 2020

PONE-D-19-29622R1

Self-determination theory in acute child and adolescent mental health inpatient care. A qualitative exploratory study.

PLOS ONE

Dear Dr. Stanton,

Thank you for submitting your manuscript to PLOS ONE. After careful consideration, we feel that it has merit but does not fully meet PLOS ONE’s publication criteria as it currently stands. Therefore, we invite you to submit a revised version of the manuscript that addresses the points raised during the review process.

Additionally, authors should add the mean and median length of interviews as requested by a reviewer. This can be calculated mathematically from the interviews. Alternatively, authors should provide an explanation for not being in possession of the raw data to derive this information.

We look forward to receiving your revised manuscript.

Kind regards,

Janhavi Ajit Vaingankar

Academic Editor

PLOS ONE

Reviewers' comments:

Reviewer's Responses to Questions

**Comments to the Author**

1. If the authors have adequately addressed your comments raised in a previous round of review and you feel that this manuscript is now acceptable for publication, you may indicate that here to bypass the “Comments to the Author” section, enter your conflict of interest statement in the “Confidential to Editor” section, and submit your "Accept" recommendation.

Reviewer #1: All comments have been addressed

Reviewer #3: All comments have been addressed

2. Is the manuscript technically sound, and do the data support the conclusions?

Reviewer #1: (No Response)

Reviewer #3: Yes

3. Has the statistical analysis been performed appropriately and rigorously? 

Reviewer #1: (No Response)

Reviewer #3: N/A

4. Have the authors made all data underlying the findings in their manuscript fully available?

Reviewer #1: (No Response)

Reviewer #3: No

5. Is the manuscript presented in an intelligible fashion and written in standard English?

Reviewer #1: (No Response)

Reviewer #3: Yes

6. Review Comments to the Author

Reviewer #1: (No Response)

Reviewer #3: This is an excellent piece of work that is described with great attention. I enjoyed reading the manuscript. It is informative and relatable. Authors generally seem to have followed COREQ guidelines while describing the study and its results. They have focused on relatedness, autonomy and competence from the perspectives of young people and their families. I would however like to recommend that authors expand the study aims as well as their rationale for the study design to be limited to SDT. It would be also useful to cover why an initial inductive followed by a deductive analytical approach was adopted by the authors. I gather that the intention was to understand SDT framework from the onset, hence possibly more direct and deeper information could have been collected if the interview guide was designed from a deductive or framework analysis perspective. In addition, given that these were inpatients actively seeking treatment during the course of the study and all the content is in relation to three SDT principles, authors may want to add some clinical implications of these findings and the non-SDT themes that may have been identified. Authors have addressed this briefly in the methods. However, I felt this argument could be strengthened further.

7. PLOS authors have the option to publish the peer review history of their article (what does this mean?). If published, this will include your full peer review and any attached files.

Reviewer #1: No

Reviewer #3: No

---

## [Author Response · Author response to Decision Letter 1]

21 Jul 2020

Response to reviewers

Mean and median length of interviews. 

We find this an odd request in a qualitative study like this but we have added a sentence stating, “The interviews lasted between 5 to 30 minutes with a mean length of 18.77 minutes and a median length of 14.45 minutes. These interviews generated 120 pages of transcript.” 

The other issues raised by reviewers to address were those indentified by Reviewer 3 in the following paragraph:

Reviewer #3: This is an excellent piece of work that is described with great attention. I enjoyed reading the manuscript. It is informative and relatable. Authors generally seem to have followed COREQ guidelines while describing the study and its results. They have focused on relatedness, autonomy and competence from the perspectives of young people and their families. I would however like to recommend that authors expand the study aims as well as their rationale for the study design to be limited to SDT. It would be also useful to cover why an initial inductive followed by a deductive analytical approach was adopted by the authors. I gather that the intention was to understand SDT framework from the onset, hence possibly more direct and deeper information could have been collected if the interview guide was designed from a deductive or framework analysis perspective. In addition, given that these were inpatients actively seeking treatment during the course of the study and all the content is in relation to three SDT principles, authors may want to add some clinical implications of these findings and the non-SDT themes that may have been identified. Authors have addressed this briefly in the methods. However, I felt this argument could be strengthened further.

An initial inductive approach was used because we felt it was premature to focus the study questions solely on SDT which implied the assumption that these principles can be considered central to providing acute inpatient care. Therefore, we collected data broadly to allow the possibility that the We analyzed the data both openly and with an intentional focus on SDT. Most of what the young people spoke about did relate to the principles of SDT. We have made this clearer in the introduction to the results. With the following sentence:

“The only issues which were not able to be analysed within the SDT framework were some mixed feelings about medication and privacy and dissatisfaction with food.”

 We agree with the reviewer that it would be appropriate to build on the findings of this exploratory study and focus more intensively on SDT in researching inpatient units.

---

## [Decision Letter · Decision Letter 2]

21 Aug 2020

PONE-D-19-29622R2

Self-determination theory in acute child and adolescent mental health inpatient care. A qualitative exploratory study.

PLOS ONE

Dear Dr. Stanton,

Thank you for submitting your manuscript to PLOS ONE. After careful consideration, we feel that it has merit but does not fully meet PLOS ONE’s publication criteria as it currently stands. Therefore, we invite you to submit a revised version of the manuscript that addresses the points raised during the review process.

We look forward to receiving your revised manuscript.

Kind regards,

Janhavi Ajit Vaingankar

Academic Editor

PLOS ONE

Reviewers' comments:

Reviewer's Responses to Questions

**Comments to the Author**

1. If the authors have adequately addressed your comments raised in a previous round of review and you feel that this manuscript is now acceptable for publication, you may indicate that here to bypass the “Comments to the Author” section, enter your conflict of interest statement in the “Confidential to Editor” section, and submit your "Accept" recommendation.

Reviewer #3: All comments have been addressed

2. Is the manuscript technically sound, and do the data support the conclusions?

Reviewer #3: Yes

3. Has the statistical analysis been performed appropriately and rigorously? 

Reviewer #3: N/A

4. Have the authors made all data underlying the findings in their manuscript fully available?

Reviewer #3: No

5. Is the manuscript presented in an intelligible fashion and written in standard English?

Reviewer #3: Yes

6. Review Comments to the Author

Reviewer #3: I have no further comments except for a new observation around the additional information provided in the manuscript; relating to the length of the interviews. I see from the mean and median time (in my opinion could be given to just one decimal point), the interviews were rather short. Half of these were completed in below 15 minutes. I find that to be a major limitation. Perhaps the authors could comment about it in their limitations, providing reasons for such short interviews (clinical population, children, etc) and justify how confident they are about the richness of their qualitative content.

7. PLOS authors have the option to publish the peer review history of their article (what does this mean?). If published, this will include your full peer review and any attached files.

Reviewer #3: No

---

## [Author Response · Author response to Decision Letter 2]

2 Sep 2020

There is one comment to address.

Reviewer #3: I have no further comments except for a new observation around the additional information provided in the manuscript; relating to the length of the interviews. I see from the mean and median time (in my opinion could be given to just one decimal point), the interviews were rather short. Half of these were completed in below 15 minutes. I find that to be a major limitation. Perhaps the authors could comment about it in their limitations, providing reasons for such short interviews (clinical population, children, etc) and justify how confident they are about the richness of their qualitative content.

On a previous review Reviewer #3 had requested the mean and median of the interview length. I thought this was unusual in a qualitative study but included them, to two decimal places, which as Reviewer #3 says, made no sense. Mean and median figures do not represent such a small number of interviews well, particularly given that there are two different sets of interviews. We have included a narrative description of the interviews which represents them better and included a statement about short interviews in the limitations section.

“Seven of the initial face to face interviews done in the unit were around 20 – 30 minutes with the rest ranging in length down to 5 minutes. Young people engaged in these to a variable degree. Some were difficult to engage and others talked freely. The follow up interviews were all shorter, six minutes or less and engagement was limited, with three of the young people saying little more than that they had nothing to add.”

“The interviews are short which is to be expected in interviews with adolescents in acute inpatient mental health care. The difference in length and engagement in the interviews done face to face in the unit and follow up by phone supports the approach to interviewing young people while in hospital despite the challenges involved in a safe consent process for this vulnerable population. The richness of qualitative data is inevitably limited in such short interviews, but this exploratory study opens important avenues for further study.”

---

## [Editor Report · Decision Letter 3]

15 Sep 2020

Self-determination theory in acute child and adolescent mental health inpatient care. A qualitative exploratory study.

PONE-D-19-29622R3

Dear Dr. Stanton,

We’re pleased to inform you that your manuscript has been judged scientifically suitable for publication and will be formally accepted for publication once it meets all outstanding technical requirements.

Kind regards,

Janhavi Ajit Vaingankar

Academic Editor

PLOS ONE

---

## [Editor Report · Acceptance letter]

7 Oct 2020

PONE-D-19-29622R3 

Self-determination theory in acute child and adolescent mental health inpatient care. A qualitative exploratory study. 

Dear Dr. Stanton:

I'm pleased to inform you that your manuscript has been deemed suitable for publication in PLOS ONE. Congratulations! Your manuscript is now with our production department. 

Kind regards, 

on behalf of

Ms Janhavi Ajit Vaingankar 

Academic Editor

PLOS ONE